# Emerin Represses STAT3 Signaling through Nuclear Membrane-Based Spatial Control

**DOI:** 10.3390/ijms22136669

**Published:** 2021-06-22

**Authors:** Byongsun Lee, Seungjae Lee, Younggwang Lee, Yongjin Park, Jaekyung Shim

**Affiliations:** Department of Bioresources Engineering, Sejong University, Seoul 05006, Korea; coolbs@sju.ac.kr (B.L.); dltmdwo920@naver.com (S.L.); shfdlxj31@naver.com (Y.L.); yongjinp3@gmail.com (Y.P.)

**Keywords:** JAK, STAT3, emerin, muscular dystrophy

## Abstract

Emerin is the inner nuclear membrane protein involved in maintaining the mechanical integrity of the nuclear membrane. Mutations in EMD encoding emerin cause Emery–Dreifuss muscular dystrophy (EDMD). Evidence is accumulating that emerin regulation of specific gene expression is associated with this disease, but the exact function of emerin has not been fully elucidated. Here, we show that emerin downregulates Signal transducer and activators of transcription 3 (STAT3) signaling, activated exclusively by Janus kinase (JAK). Deletion mutation experiments show that the lamin-binding domain of emerin is essential for the inhibition of STAT3 signaling. Emerin interacts directly and co-localizes with STAT3 in the nuclear membrane. Emerin knockdown induces STAT3 target genes *Bcl2* and *Survivin* to increase cell survival signals and suppress hydrogen peroxide-induced cell death in HeLa cells. Specifically, downregulation of BAF or lamin A/C increases STAT3 signaling, suggesting that correct-localized emerin, by assembling with BAF and lamin A/C, acts as an intrinsic inhibitor against STAT3 signaling. In C2C12 cells, emerin knockdown induces STAT3 target gene, Pax7, and activated abnormal myoblast proliferation associated with muscle wasting in skeletal muscle homeostasis. Our results indicate that emerin downregulates STAT3 signaling by inducing retention of STAT3 and delaying STAT3 signaling in the nuclear membrane. This mechanism provides clues to the etiology of emerin-related muscular dystrophy and may be a new therapeutic target for treatment.

## 1. Introduction

The nuclear lamina provides mechanical assistance to the nucleus and a protein network that contributes to DNA replication, gene regulation, and genome stability [1,2]. The nuclear lamina proteins scaffold hundreds of proteins, including the LAP2-emerin-MAN1-domain (LEM-D) protein family in the inner nuclear membrane [3,4]. Multiple human diseases are caused by the loss of individual nuclear lamina proteins, highlighting the importance of this network [2,5]. Emerin was initially identified as a 35 kDa inner nuclear transmembrane protein that interacts with structural proteins including lamin A/C, nesprin-1α/2β, and BAF [6,7,8,9]. In addition to its known function in supporting the mechanical integrity of the nuclear membrane, emerin is involved in sensing and responding to mechanical tension of the nuclear membrane [10]. Emerin also plays a role in the signaling and transcriptional regulation to interact many proteins, including β-catenin, notch intracellular domain (NICD), germ cell-less (GCL), Bcl-2-associated transcription factor (Btf), and LIM Domain Only 7 (LMO7) [11,12,13,14,15,16]. The mutations in the emerin network cause Emery–Dreifuss muscular dystrophy (EDMD) [2,17,18]. Recently, the nucleotide sequence of emerin mutant that induces EDMD was revealed [17]. X-linked EDMD (X-EDMD) is caused by mutations in EMD located on chromosome Xq28, and autosomal dominant EDMD (AD-EDMD) is caused by mutations in LMNA (encoding Lamin A) located on chromosome 1q11–q23 [18,19,20,21]. The loss of the emerin protein results in skeletal muscle wasting and cardiac defects, which characterize EDMD [22,23]; however, the role of emerin loss in this disease has not been precisely elucidated.

In the present study, we assessed the effect of emerin on gene expression using a transcription factor profiling PCR array covering 84 genes. Based on the positive value of transcription factor profiling, we tried to confirm the possibility of STAT3 signaling as a target pathway regulated by emerin. STAT3, a member of the STAT protein family, was first discovered in interferon (IFN) signaling studies [24,25,26]. STAT3 is expressed at a low basal level in virtually all cells. A variety of stimuli can activate STAT3 signaling via phosphorylation of tyrosine 705. In general, cytokines such as interleukin-6 (IL-6) [27], leukemia inhibitory factor (LIF) [28], IL-5 [29], IFN-γ [30], and TNF-α [31] are known to activate STAT3. In addition to cytokines, growth factors such as EGF [32], TGF α [31], and PDGF [33] activate STAT3. In response to cytokines and growth factors, STAT3 is phosphorylated by receptor-associated JAK, forms homo- or heterodimers, and translocates to the cell nucleus where it acts as a transcription activator [34]. Additionally, activation of STAT3 may occur via phosphorylation of serine 727 by mitogen-activated protein kinase (MAPK) [35] and through c-Src non-receptor tyrosine kinase [36,37].

Among the various roles of STAT3, many downstream genes are expressed in cell survival [38,39,40,41], cell proliferation [42,43,44], inflammation [27,45], and tumorigenesis [46,47,48,49,50,51]. STAT3 signaling plays a critical role in muscle wasting induced by the IL6/JAK/STAT3 signaling pathway [52,53]. STAT3 signaling is activated in skeletal muscle and promotes skeletal muscle atrophy in muscle diseases, such as Duchenne muscular dystrophy (DMD) and merosin-deficient congenital muscular dystrophy (MDC1A) [54,55,56]; thus, prolonged activation of STAT3 in muscles is responsible for muscle wasting by activating protein degradation pathways. It is important to balance the extent of STAT3 activation and the duration and location (cell types) of STAT3 signaling when developing therapeutic interventions [57].

In the study, we examined the role of emerin as a transcriptional inhibitor of STAT3 target genes, such as cell survival-related genes including *Bcl2* and *Survivin* in HeLa cells. We found that emerin can modulate STAT3 signaling by inducing the retention of the STAT3 at the inner nuclear membrane. Cytotoxicity studies using emerin knockdown HeLa cells confirmed that STAT3 signaling induced STAT3 target genes *Bcl2* and *Survivin* to increase cell survival signals and suppress hydrogen-peroxide-induced cell death. Using mouse myoblast C2C12 cells, we found that emerin can modulate myoblast proliferation by inhibiting STAT3 signaling. These findings imply that normal STAT3 activity under emerin regulation is required for proper muscle maintenance. However, intermittent STAT3 inhibition may have promising implications for increasing muscle regeneration in emerin-related muscular dystrophy. All our results suggest that misregulation of the STAT3 signaling pathway, which is essential for skeletal muscle development, may lead to LEM-D protein-associated human diseases such as EMDM.

## 2. Results

### 2.1. Emerin Represses STAT3 Transcriptional Activity

When the STAT3 signal, known to be essential for cancer cell proliferation, is activated, cytoplasmic STAT3 protein moves to the nucleus through the nuclear inner membrane where emerin is present. In addition, related studies have shown that emerin regulates Notch and Wnt signaling [11,15]. Here, we investigated the involvement of emerin in STAT3 signaling in the same manner as a previously reported human transcription factor profiling assay that measures the expression levels of genes using qRT-PCR [15]. We analyzed the expression of 84 genes affected by emerin and present the results in Appendix A. Interestingly, we found that the expression of the STAT family was significantly reduced by emerin, and MyoD and Myf5, which are related to muscle development, were reduced. STAT-signal-related genes (red) and muscle development-related genes (blue) are marked differently with the 84 gene rankings in Appendix A.

Since the previously reported emerin-regulated *Notch* gene was not included in the 84 genes, we conducted a separate experiment to confirm the importance of performing transcription factor profiling analysis. Reduction in STAT3 expression by overexpressed emerin was confirmed in the same pattern as decreasing Notch expression (Figure 1A). In addition, our results showed that emerin suppresses the expression of cyclinD1, the β-catenin target gene in Wnt signaling (Figure 1A). Conversely, the knockdown of emerin by siRNA led to upregulation of *STAT3*, *Notch*, and *cyclinD1* (Figure 1B). These results are consistent with those of previous studies [11,15], indicating that our experimental method for testing the effect of emerin on STAT3 is appropriate. Furthermore, this experiment showed that STAT3 signaling regulation by emerin is similar to controlling Notch signaling through the spatial control of emerin in the nuclear membrane [15].

Additionally, we examined whether emerin affects the expressions of *Bcl2* and *Survivin*, survival-related genes downstream STAT3. Emerin inhibited mRNA expression of *Bcl2* and *Survivin* (Figure 1C). Similar to Figure 1B, emerin decreased by siRNA increased *Bcl2* and *Survivin* of the STAT3 target gene (Figure 1D). We performed Western blot experiments to confirm the protein expression of the representative target genes (Notch and Bcl2) along with STAT3, separate from emerin RNA interference experiments. The Western blot results are consistent with the gene expression levels resulting from si-Emerin treatment (Figure 1F).

To determine if STAT3 is be a direct target of emerin, we built a STAT3-luciferase assay system on HeLa cell double-infected STAT3-luciferase particles and Renilla particles. Treatment with IL-6 as a STAT3 activation signal in HeLa-STAT3-Luc/Ren cells increased STAT3 transcriptional activity as indicated by an increased amount of IL-6 (Figure 1E). In a STAT3-luciferase reporter experiment, we confirmed that IL-6 activated STAT3 transcriptional activity and that activated STAT3 was downregulated by emerin (Figure 1E). Interestingly, the decrease in the transcriptional activity of STAT3 in the nucleus, which is dependent on emerin, resulted in a significant reduction in phosphorylated STAT3, an activated form completed in the cytoplasm (Figure 1G). This means that emerin may negatively affect the expression or activation process of JAK or MAPK, which are involved in STAT3 activation. A study related to our results reported the general activation of the MAPK pathway in EMD null mice, an EMDM model [58].

In particular, the graph showing the relative expression levels of STAT3 indicates the difference in the level of STAT3 at 15 min after IL-6 treatment and the phosphorylation of STAT3 30 min after IL-6 treatment (Figure 1G).

### 2.2. Emerin Interacts with STAT3 Proteins

Emerin is known to form a network in the nuclear membrane with several proteins including lamin A/C and BAF. Emerin regulates gene expression by interacting with several proteins while maintaining the integrity of the nuclear membrane [9,59]. Emerin may regulation of STAT3 transcriptional activity through two possibilities: through direct interaction and through other proteins in the network. We performed a co-immunoprecipitation (Co-IP) experiment to determine whether direct interaction with emerin occurs. Co-IP experiments showed that emerin and STAT3 can bind directly at the endogenous level. The interaction between the two proteins was not related to the presence or absence of phosphorylation of STAT3 (Figure 2A). Emerin, which is present in the nuclear membrane, is expected to bind to activated STAT3 migrating to the nucleus and to regulate pSTAT3 transcriptional activity. Immunocytochemistry (ICC) experiments showed that activated STAT3 migrates to the nucleus, and some pSTAT3 is located in the nuclear membrane with emerin. The orange color in the merged figure shows that emerin (red) and pSTAT3 (green) can interact in the same location of the nuclear membrane (enlarged merge in Figure 2B).

We also investigated which part of emerin is essential in binding to STAT3 using the overexpressed deletion mutation of emerin in Co-IP experiments. Co-IP data showed that most of the mutations can bind in different degrees to STAT3 and the D5 mutant has the weakest binding. Co-IP data showed that trans-membrane (TM), a moiety required for the anchorage of emerin, and the LEM domain that binds BAF are not essential for binding. Instead, the lamin-binding (LB) domain binds to STAT3, as all deletion mutants have LB domains in common.

We tried to determine which emerin deletion mutations affect STAT3 transcriptional inhibition through a STAT3 luciferase assay. As shown in Figure 1E, IL-6-induced STAT3 transcriptional activity was inhibited by emerin, and the emerin deletion mutants D2 and D3, containing TM domain and LB domain, had a significant inhibitory effect on STAT3 transcriptional activity compared to wild-type emerin (Figure 2D). However, emerin mutants D1 and D4, capable of binding to STAT3 (Figure 2C), did not show inhibition of transcriptional activity (Figure 2D), indicating that STAT3 transcriptional activity is inhibited through emerin–STAT3 binding only when emerin is anchored to the nuclear membrane. These data suggest that the localization of emerin proteins to the inner nuclear membrane is essential for suppressing STAT3 signaling.

### 2.3. Emerin-Network Affects Cell Survival Through Modulating STAT3 Signal

Emerin predominantly localizes to the inner nuclear membrane by binding to A-type lamins (nuclear intermediate filament proteins) and a chromatin protein, BAF [60]. BAF is essential for the localization of emerin to the nuclear envelope because, in the absence of BAF, emerin is sequestered in the cytosol [59,61]. Thus, loss of BAF and lamin A/C may increase STAT3 signaling by inducing the mislocalization of emerin to the cytosol. Therefore, we studied changes in the transcriptional activity of STAT3 when the emerin network was disrupted by siRNA, including si-Lamin A/C and si-BAF.

An increase in STAT3 expression confirmed the effect of the disrupted emerin network through the same qRT-PCR experiment as in Figure 1D, in which si-Emerin was introduced (Figure 3A). In addition, like the emerin knockdown experiment, *Bcl2* and *Survivin* expression increased in Hela cells using si-BAF and si-Lamin A/C (Figure 3A). These results showed that if the protein constituting the emerin network is impaired, the entire emerin network is also defective. A defective emerin network reduces the emerin inhibitory effect on STAT3 transcriptional activity, increasing the expression of STAT3, thereby increasing the expressions of *Bcl2* and *Survivin*, known as anti-apoptotic genes [62,63].

To investigate the relationship between the expression levels of *Bcl2* and *Survivin* downstream of STAT3 and cell survival, HeLa cells were treated with H_2_O_2_ to induce cell death under the reduction in emerin localization in the nuclear inner membrane using si-Emerin, si-BAF, and si-Lamin A/C. Under these circumstances, IL-6 treatment appeared to inhibit H_2_O_2_-induced cell death by significantly increasing STAT3 signaling activation (Figure 3C). Furthermore, the emerin network downregulation using siRNA increased cell viability even without IL-6 treatment (Figure 3B). The increased expression of STAT3 increasing the expression of the anti-apoptotic genes *Bcl2* and *Survivin* was expected to sufficiently delay H_2_O_2_-induced apoptosis.

These results suggest that emerin plays a role in survival through, at least in part, the modulation of STAT3 signaling in HeLa cells. Using a simple schematic, we present a model demonstrating the potential of emerin to modulate JAK-STAT signaling in the fifth figure in the main text. The schematic model is explained as follows: IL-6 activates the STAT3 present in the cytoplasm to form the JAK–STAT3 complex. Activated pSTAT3 enters the nucleus and acts as a transcriptional activator for target genes. Thus, the inner nuclear emerin network including lamin A/C, BAF, and emerin regulates transcription factors such as STAT3 that enter the inner nuclear membrane.

### 2.4. Emerin Regulates Muscle Cell Proliferation Through STAT3 Signaling

Emerin and STAT3 are well-known to play essential roles in the etiology of muscle disease [18,57]. Therefore, we used C2C12 myoblast cells in our experiments to investigate the regulatory effect by emerin of STAT3 on muscle cells. In C2C12 cells, knockdown of emerin increased STAT3 mRNA, similar to the results in HeLa cells, and increased the PAX7 of the myoblast proliferation gene (Figure 4A). After reduction in emerin using si-Emerin in C2C12 cells, PAX7, MyoD, and Myogenin (MyoG) expression levels were checked for three days in differentiation medium (DM). Emerin knockdown in DM-treated C2C12 cells increased STAT3 mRNA and increased the PAX7 of the myoblast proliferation gene, similar to the results of the standard medium in Figure 4A. Conversely, MyoD and MyoG, the myocyte differentiation genes, decreased (Figure 4B). In addition, as a result of observation for 2 days after si-Emerin treatment, the growth in C2C12 cells was promoted compared to the control, probably due to the increased STAT3 target gene, PAX7 (Figure 4C).

As in previous studies [56,64], increasing STAT3 inhibited muscle cell differentiation and promoted muscle cell proliferation by emerin knockdown. Therefore, we investigated the physiological changes due to the regulation of STAT3 by emerin in C2C12 cell differentiation. In Figure 4B, Myosin Heavy Chain (MYHC) decreased with STAT3 increased by si-Emerin on day 3 of DM in the ICC of MYHC (Figure 4D). However, on the fifth day of DM, as the knockdown effect by si-Emerin decreased, there was no significant increase in PAX7 or decrease in MyoD and MyoG, and there was not much difference in the level of MYHC expression in the ICC results (Appendix A). These results suggest that emerin plays a role in muscle cell development by engaging in muscle cell proliferation through modulation of STAT3 signaling. We express these proposals using a simple schematic model in the fifth figure in the main text. Cytoplasmic pSTAT3 activated by various signals enters the nucleus and regulates genes’ expressions in proliferation and differentiation for muscle maintenance. At this time, the inner nuclear emerin network containing emerin properly regulates the transcriptional activity of pSTAT3 to maintain normal muscle homeostasis. Conversely, activated STATs that are not under the control of the emerin network lead to increased expression of genes associated with muscle cell proliferation, and this imbalance may lead to myofiber regeneration problems such as muscular dystrophy.

## 3. Discussion

Skeletal muscle development is a complex regulatory process that fine-tunes the balance of catabolic and anabolic processes regulated by muscle genes and transcription factors. Depending on the stringent temporal sequence expression, these transcription factors precisely control the proliferation, differentiation, and fusion of muscle cells [64,65,66]. However, muscular dystrophy impairs the microscopic homeostasis, which slows muscle generation and repair, leading to an increased catabolic process, resulting in the loss of skeletal muscle mass by increasing proteolysis through the ubiquitin-proteasome autophagy-lysosome system [57]. Thus, representative muscular dystrophy, X-EMDM, is presumed to cause problems in several biological processes due to the lack of emerin.

In the present study, we performed transcription profiling analysis to explore the role of emerin in gene regulation. We found that emerin plays a role in the transcriptional suppression of many genes, including those of the Notch, Wnt, and STAT signaling pathways (Figure 1A,B and Appendix A). These results are consistent with previously reported results because emerin can negatively regulate gene expression through the recruitment of transcription inhibitors. For example, the direct interaction of emerin and β-catenin triggers the downregulation of Wnt signaling [11]. Conversely, in the absence of emerin, the level of nuclear β-catenin increases, leading to upregulation of the target gene [11]. In addition, emerin induces nuclear envelope localization of LMO7, a transcriptional activator for muscle differentiation, and inhibits transcriptional function [13]. Our transcription profiling analysis showed significant relevance to previous studies related to myogenesis, such as the temporal balance between Notch and Wnt signaling orchestrating the precise progression of muscle precursor cells along the myogenic lineage pathway [67].

Transcription factor profiling analysis also showed that the level of expression of STAT families by emerin is generally low (Appendix A). We were able to confirm once again that it inhibits STAT3 transcriptional activity in HeLa cells (Figure 1C,E). STAT3 signaling has been reported to play a critical role in satellite cell myogenic capacity and self-renewal [68,69]. In addition, activated STAT3 promotes skeletal muscle atrophy in muscle diseases [54] while acting as an oncogene in non-muscular cells [70,71]. An exciting aspect of the transcription profiling analysis results is that in addition to significantly reducing the expression of stat3, overexpression of emerin also decreased the expression of MyoD and MYF5, which are involved in skeletal muscle development. This result was from HeLa cells in a typical medium that induces cell proliferation (Appendix A). However, even when STAT3 expression was increased in emerin knockdown C2C12 myoblasts using DM, which promotes differentiation, the expressions of MyoD and MyoG decreased (Figure 4B). This finding suggests that even muscle differentiation is related to STAT3 released from emerin, causing upregulation of Pax7, promoting cell proliferation (Figure 5). All these findings reinforce the suggestion that LEM-D protein-associated human diseases such as EMDM are associated with the misregulation of signaling pathways essential for skeletal muscle cell proliferation and differentiation.

Mutations in the genes encoding emerin (*EMD*) and lamin A/C (*LMNA*) can cause different forms of EDMD. The functional link between the two proteins and both being implicated in very similar diseases suggest that they work through a convergent pathway. Previous reports have shown that the EDMD model, mouse embryonic fibroblasts (MEFs) from *Lmna*^−/−^ mice, has a fast-growing phenotype similar to emerin null fibroblasts [72]. We found that the loss of lamin A/C and BAF leads to erroneous localization and the loss of function of emerin in the cytoplasm [15], so upregulated STAT3 signaling may contribute to cell survival in HeLa cells. Based on these results, we suggest that the cause of each EDMD in different gene mutations is due to the same expansion that forms a proliferating myoblasts pool by upregulated STAT3 signaling. In this regard, we showed that emerin can downregulate myoblast proliferation by inhibiting STAT3 signaling in C2C12 cells (Figure 4). As shown in the emerin-STAT3 model in Figure 5, STAT3 activity attenuated by interacting emerin in the inner nuclear membrane is sufficient to fulfill its essential functions in gene expression regulation during myoblast differentiation, thus demonstrating the importance of emerin in myogenesis.

In conclusion, our results suggest that defects in the regulation of muscle cell proliferation due to incorrect regulation of STAT3 signaling may be one cause of EDMD pathology.

## 4. Materials and Methods

### 4.1. Antibodies and Chemicals

For immunoblotting and immunocytochemistry, primary antibodies for emerin (#sc-15378, 1:4500), Lamin A/C (#sc-20681, 1:4500), BAF (#sc-166324, 1:5000), Bcl2 (#sc-7382, 1:4500), and GAPDH (#sc-47724, 1:5500) were purchased from Santa Cruz (Santa Cruz Biotechnology, Santa Cruz, CA, USA). Primary antibodies specific for pSTAT3 (#ab76315, 1:3000) were purchased from Abcam (Cambridge, U.K.). Primary antibodies specific for HA (1:5000), Flag (1:5000), and β-actin (1:5000) were purchased from MBL (Woburn, MA, USA). Primary antibodies specific for STAT3 (1:3000) were purchased from Cell Signaling (Danvers, MA, USA). Primary antibodies for Myosin Heavy Chain (MYHC, #MAB4470, 1:50) were purchased from R&D systems (Minneapolis, MN, USA). For the viability assay, cells were treated H_2_O_2_ for cell death, and thiazolyl blue tetrazolium bromide (MTT) was purchased from Bio Basic (Markham, ON, Canada).

### 4.2. Cell Culture and Transfection

HeLa cells (American Type Culture Collection; ATCC) were cultured in DMEM (Biowest, Riverside, MO, USA) supplemented with 10% FBS (Welgene, South Korea) and 1% antibiotic antimycotic solution, 100× (Corning, Manassas, VA, USA). Transfection was executed using Lipofectamine 2000 reagent (Invitrogen, Grand Island, NY, USA) according to the manufacturer’s instructions. The transfected cells were cultured for 24–48 h, washed with DPBS, and harvested with lysis buffer (Invitrogen, Grand Island, NY, USA). C2C12 cells were a gift from Yong-Gyu Ko at Korea University in South Korea. C2C12 cells were cultured in DMEM supplemented with 10% FBS and 1% antibiotic antimycotic solution, 100× (Biowest, Riverside, MO, USA). For induction of differentiation, fully confluent C2C12 cells were cultured in DMEM supplemented with 2% horse serum (Sigma-Aldrich, MO, USA) for 3 days.

### 4.3. Plasmid Constructs

The human *emerin* cDNA (NM_000117) was provided by the 21C Frontier Human Gene Bank, Seoul, South Korea, amplified by PCR, and inserted into the restriction enzyme sites (BamHI and XhoI) of pcDNA3-HA for biochemical studies. For construction of *emerin* deletion mutants, the corresponding regions were amplified by PCR and inserted into the restriction enzyme sites (BamHI and XhoI) of pcDNA3-HA and pcDNA3-Flag. The *emerin* deletion mutants cloning primer sequences are: WT (forward 5′-CGGGATCCCGAAATGGACAACTACGCAGAT-3′, reverse 5′-CCGCTCGAGCGGCTAGAAGGGGTTGCCTTC-3′), D1 (forward 5′-CGGGATCCCGAAATGGACAACTACGCAGAT-3′, reverse 5′-CCGCTCGAGCGGCTAGTTTTCAGGCCGGATGGC-3′), D2 (forward 5′-CGGGATCCCCGAAATGGAGAAGAAGATCTTCGAG-3′, reverse 5′-CCGGTCGAGCGGCTAGAAGGGGTTGCCTTC-3′), D3 (forward 5′-CGGGATCCCGAAATGCTCTACCAGAGCAAGGGC-3′, reverse 5′-CCGGTCGAGCGGCTAGAAGGGGTTGCCTTC-3′), D4 (forward 5′-CGGGATCCCGAAATGGACAACTACGCAGAT-3′, reverse 5′-CCGGTCGAGCGGCTAGGAGGCTGAAACAGGGCG-3′), D5 (forward 5′-CGGGATCCCGAAATGGAGAAGAAGATCTTCGAG-3′, reverse 5′-CCGCTCGAGCGGCTAGTTTTCAGGCCGGATGGC-3′). Human *STAT3* (NM_003150) cDNA was provided by 21C Frontier Human Gene Bank (South Korea), amplified by PCR, and inserted into the restriction enzyme sites (HindIII and SalI) of pcDNA3-Flag. The STAT3 cloning primer sequences are: WT (forward 5′-AAGCTTATGGCCCAATGGAATCAGCTA-3′; reverse 5′-GTCGACTCACATGGGGGAGGTAGCGCA-3′).

### 4.4. Transcription Factor Profiling Assay

Total RNA was isolated using Trizol reagent (Life Technologies, Carlsbad, CA, USA), and 1 μg of total RNA was used for cDNA synthesis. The human transcription factor profiling PCR array was performed according to the manufacturer’s protocol (#PAHS-075ZC-2; Qiagen, Valencia, CA, USA). Data were obtained following the manufacturer’s instructions.

### 4.5. Luciferase Reporter Assay

HeLa cells were dual-infected with STAT3 lentiviral luciferase and Renilla lentiviral luciferase (#CCS-9028L, Qiagen, Hilden, Germany), selected by puromycin (stable cell line-Hela STAT3 Luc/Ren). Cells were grown to 70–80% confluence in 6-well plates and transiently transfected with encoding emerin and incubated for 24 h at 37 °C in a CO_2_ incubator. Then, cells were treated with IL-6 for 6 h, lysed in 5× passive lysis buffer (Promega, Madison, WI, USA), and analyzed for dual luciferase activity with a GloMax^®^ 96-well microplate luminometer system (Promega). The luciferase reporter activity in each sample was normalized to Renilla protein activity (Promega).

### 4.6. Immunoblotting

For Western blot analysis, all proteins were separated by SDS-PAGE polyacrylamide gel electrophoresis and transferred on polyvinylidene difluoride (PVDF) membranes (Millipore, Billerica, MA, USA). Membranes were blocked for 1 h at RT with a solution of 4% non-fat milk powder or 4% BSA in TBS containing 0.05% Tween-20 (TBST). The membranes were incubated with the first antibody in blocking solution ON at 4 °C. The membranes were washed 3 times with TBST and incubated with the second antibody for 2 h at RT. After washing three times with TBST, the membranes were developed using an electrochemiluminescence (ECL) detection system (Bio-Rad, Hercules, CA, USA).

### 4.7. Immunoprecipitation

The cells were scraped off the culture dish with 1000 μL cold RIPA buffer (Sigma-Aldrich, MO, USA) containing protease inhibitor cocktail solution (GenDEPOT, Barker, TX, USA). The cell lysate was centrifuged at 13,000 rpm for 15 min at 4 °C. The supernatant was harvested, and its protein concentration was measured with a BCA protein assay kit (Thermo Scientific). For immunoprecipitation, 300 μg protein in RIPA buffer was incubated with 2 μg antibody per 300 μg total protein for overnight at 4 °C with constant rotation. Then, a pre-mixture of protein and antibody was incubated with 25 μL of a 50% slurry of protein-G-Agarose beads (Amicogen, South Korea) for 2 h at 4 °C under constant rotation. After centrifugation at 1000 rpm for 1 min at 4 °C, the supernatant was discarded, and the agarose-bead pellets were washed with PBS. After a final wash, the agarose-bead pellets were resuspended in a final volume of 20 μL with sample buffer (50 mM Tris–HCl pH 6.8, 2% SDS, 10% glycerol, 1% β-mercaptoethanol, 12.5 mM EDTA, and 0.02% bromophenol blue) and boiled at 90 °C for 5 min. Then, the agarose beads were removed by brief centrifugation, and the supernatant was loaded immediately onto an 8–10% SDS-polyacrylamide reducing gel. As a next step, Western blotting was performed as described above.

### 4.8. Immunocytochemistry (ICC)

HeLa cells were plated on glass cover slips and then transfected with 1 μg of vector. After incubation for 24–48 h, then cells were fixed with 4% paraformaldehyde (PFA), permeabilized with 0.1% Triton X-100 in PBS, and then incubated with blocking Sol. After incubation ON with the first HA antibody (1:200) in blocking solution, cells were washed and incubated with TRITC or FITC-conjugated 2nd antibodies (1:200) for 1 h at RT. After staining with DAPI (Life technology, Carlsbad, CA, USA), cells were observed under a microscope.

### 4.9. Quantitative RT-PCR (qRT-PCR)

Total RNA was separated using the RiboEx (Gene All, South Korea), and 1 μg of total RNA was used for cDNA synthesis (Intron Bio, South Korea). cDNA was amplified using primer pairs for human *STAT3* (forward 5′-TTGACAAAGACTCTGGGGAC-3′ and reverse 5′-CAGGGAAGCATCACAATTGG-3′), human *Notch1* (forward 5′-TACGTGTGCACCTGCCGGG-3′ and reverse 5′-CGTTTCTGCAGGGGCTGGGG-3′), human *Survivin* (forward 5′-CCTTCACATCTGTCACGTTC-3′ and reverse 5′-GAAGCTGTAACAATCCACCC-3′), human *Bcl2* (forward 5′-GTGGCCTCTAAGATGAAGGA-3′ and reverse 5′-TGCGGATGATCTGTTTGTTC-3′), human *Cyclin D1* (forward 5′-GGATTGTGGCCTTCTTTGAGGA-3′ and reverse 5′-AGGTACTCAGTCATCCACAG-3′), human *GAPDH* (forward 5′-AAAAGCAGCCCTGGTGACC-3′ and reverse 5′-GTCGGAGTCAACGGATTTGG-3′), mouse *emerin* (forward 5′-CCACCAAGACATACGGGGAG-3′ and reverse 5′-TCTTGGCCATAGATGAGGCG-3′), mouse *PAX7* (forward 5′-CGACTCTGGATTCGTCTCC-3′ and reverse 5′-GGCCTTGGCCAAGAGGG-3′), mouse *MyoD* (forward 5′-GGCTACGACACCGCCTACTAC-3′ and reverse 5′-GGTCTGGGTTCCCTGTTCTG-3′), mouse *Myogenin* (*MyoG*) (forward 5′-CCTGGAAGAAAAGGGACTGG-3′ and reverse 5′-CGCTCAATGTACTGGATGGC-3′), mouse *STAT3* (forward 5′-AGGAGTCTAACAACGGCAGCCT-4′ and reverse 5′-GTGGTACACCTCAGTCTCGAAG-3′), and mouse *β-Actin* (forward 5′-GGCTGTATTCCCCTCCATCG-3′ and reverse 5′-CCAGTTGGTAACAATGCCATGT-4′). qRT-PCR was performed using the StepOne Real-Time PCR System. Reactions were amplified using the primers described above and a HiPi Real-Time PCR 2× Master MIX-SYBR green (ELPis, Daejeon, South Korea) according to the manufacturer’s instructions.

### 4.10. Small Interfering RNA

Negative siRNA was non-targeting siRNA for human, rat, and mouse. The human *emerin* (#1047199), mouse *emerin* (#13726-1), human *BAF* (#1011344), human *Lamin A/C* (#1033333), mouse *Lamin A/C* (#SP-4002), and negative control (#SN-1002) oligo siRNA were purchased from Bioneer (South Korea). Transfection was performed with Lipofectamine 2000 in HeLa and C2C12 cells according to the manufacturer’s protocol. The sequences for siRNA targeting were as follows: human *emerin* (sense sequence 5′-AGUCGAAUUCAAGUCAGAG-3′, and antisense sequence, 5′-CUCUGACUUGAAUUCACU-3′), mouse *emerin* (sense sequence 5′-GAGCAAGGACUAUAAUGA-3′, and antisense sequence 5′-AUCAUUAUAGUCCUUGCUC-3′), human *BAF* (sense sequence 5′ACUUCUGCCAGUUAGAGC-3′, and antisense sequence, 5′-GCUCUAACUGGCUAGAAGU-3′), and human *Lamin A/C* (sense sequence, 5′-CUCUGACUUGAAUUCCU-3′ and antisense sequence, 5′-AGUCGAAUUCAAGUCAGAG-3′).

### 4.11. Statistical Analysis

All results are presented as the mean ± SD. Statistical significance was determined with the Student’s *t*-test with a significance level of *p* < 0.05. The data for the transcription qRT-PCR array, dual-luciferase assay, and viability assay are presented as the mean of three independent experiments.

## Figures and Tables

**Figure 1 ijms-22-06669-f001:**
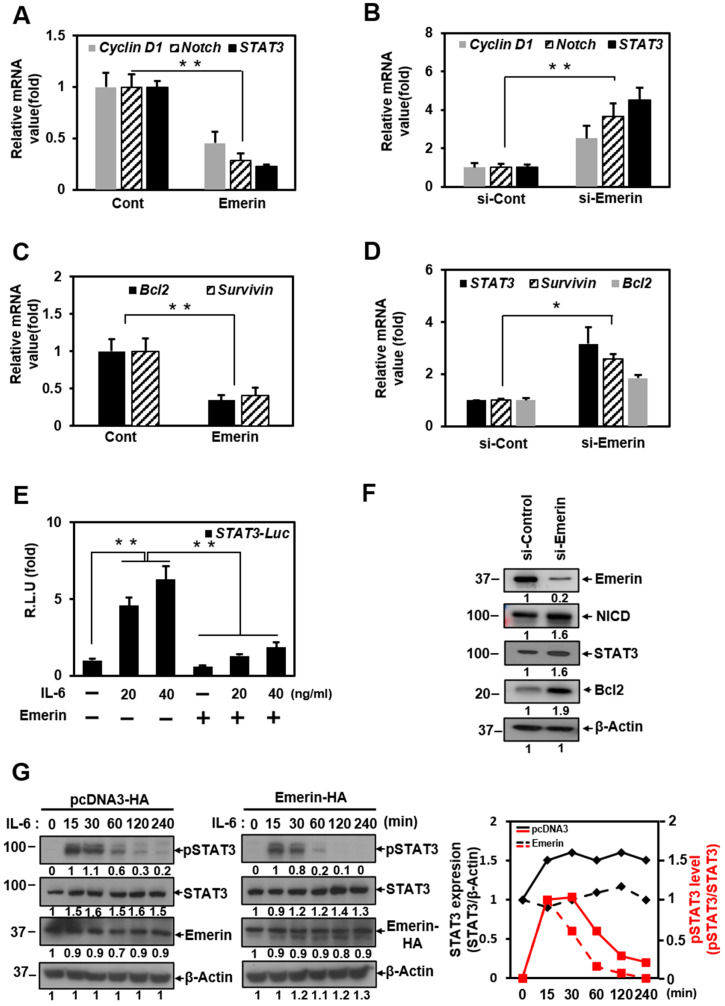
Emerin regulates the expression of STAT3 genes. (**A**,**C**) HeLa cells were transfected with vectors encoding *Emerin-HA* (1 μg) in 6-well plates for 24 h. Total RNA was isolated and subjected to qRT-PCR analysis. Data were normalized to GAPDH expression. The results are presented the mean ± SD of three independent experiments performed in triplicate. ** *p* < 0.01. (**B**,**D**) HeLa cells were treated with siRNA (100 nM) against *Emerin* or control (si-Cont) for 48 h in 6-well plates. Total RNA was isolated and subjected to qRT-PCR analysis. Data were normalized to GAPDH expression. The results are presented as the mean ± SD of three independent experiments performed in triplicate. * *p* < 0.05, ** *p* < 0.01. (**E**) HeLa cells were dual-infected with STAT3 lentiviral luciferase and Renilla lentiviral luciferase. Cells were selected by puromycin. HeLa cells were treated with increasing amounts of IL-6 (0–40 ng/mL) for 6 h before being transfected with vectors encoding *Emerin-HA* (1 μg) for 24 h in 6-well plates. Cells were lysed and subject to a luciferase assay. The luciferase reporter activity in each sample was normalized to a Renilla protein activity. R.L.U., relative luciferase units. The results are the mean ± SD of three independent experiments performed in triplicate. ** *p* < 0.01. (**F**) HeLa cells were treated with siRNA (100 nM) against *Emerin* or control (si-Cont) for 48 h in 6-well plates. Treated cells were lysed and subjected to Western blotting with antibody against emerin, NICD, STAT3, Bcl2, and β-actin. The membranes were analyzed using ImageJ software (NIH, Bethesda, NY, USA). (**G**) We treated 50% confluent HeLa cells with IL-6 (40 ng/mL) for the indicated times in 6-well plates. Treated cells were lysed and subjected to Western blotting with antibodies against STAT3, p-STAT3, emerin, HA-epitope, and β-actin. The membranes were analyzed using ImageJ software (NIH, Bethesda, NY, USA). Intensities of STAT3 and pSTAT3 in control vector transfection (left panel) and Emerin-HA transfection (right panel) were analyzed 16 h after transfection, including IL-6 treatment times.

**Figure 2 ijms-22-06669-f002:**
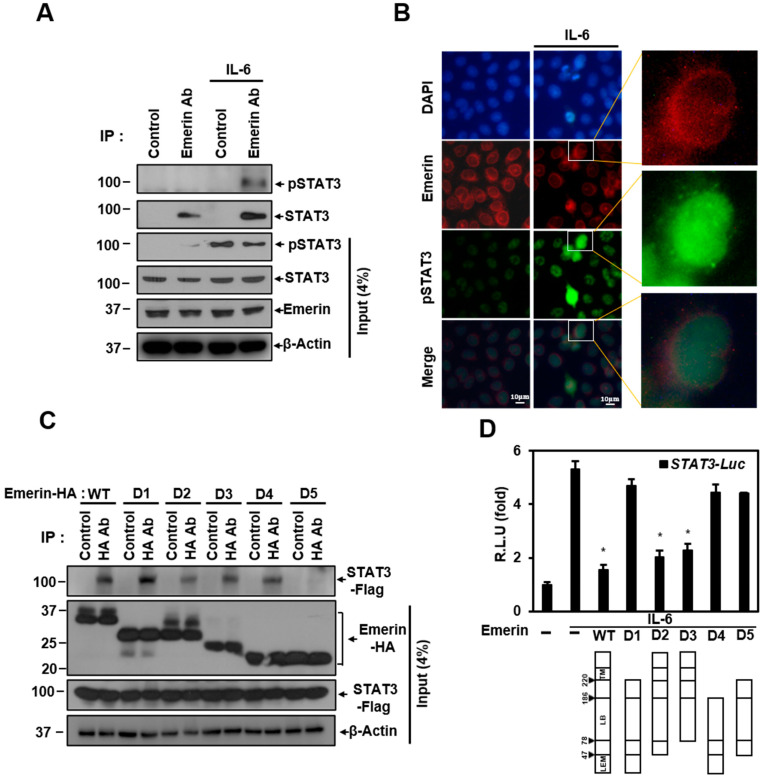
Emerin interacts with STAT3. (**A**) Confluent HeLa cells in 100 mm plates with or without IL-6 (40 ng/mL) were lysed and immunoprecipitated with antibodies against emerin. The precipitates were subjected to Western blotting with antibodies against STAT3 or p-STAT3. (**B**) Immunocytochemistry image stained with anti-pSTAT3 and emerin antibody from confluent HeLa cells on a coverslip on 30 mm plates with or without IL-6 (40 ng/mL). DAPI (blue) was used to visualize the nucleus. Scale bar = 10 μm. (**C**) HeLa cells were transiently co-transfected with vectors encoding *HA-Emerin* deletion mutants (6 μg) and *Flag-STAT3* (6 μg) in 100 mm plates for 24 h. Cells were lysed and immunoprecipitated with antibodies against HA for overnight at 4 °C, and the precipitates were subjected to Western blotting with antibodies against Flag. (**D**) Dual-infected Hela cells were transiently transfected with vectors encoding *Emerin* deletion mutants (1 μg) for 24 h and treated for 6 h with IL-6 (40 ng/mL) in 6-well plates. Cells were lysed and subjected to a luciferase assay. The luciferase reporter activity in each sample was normalized to Renilla protein activity. R.L.U., relative luciferase units. The results are the mean ± SD of three independent experiments performed in triplicate. * *p* < 0.05.

**Figure 3 ijms-22-06669-f003:**
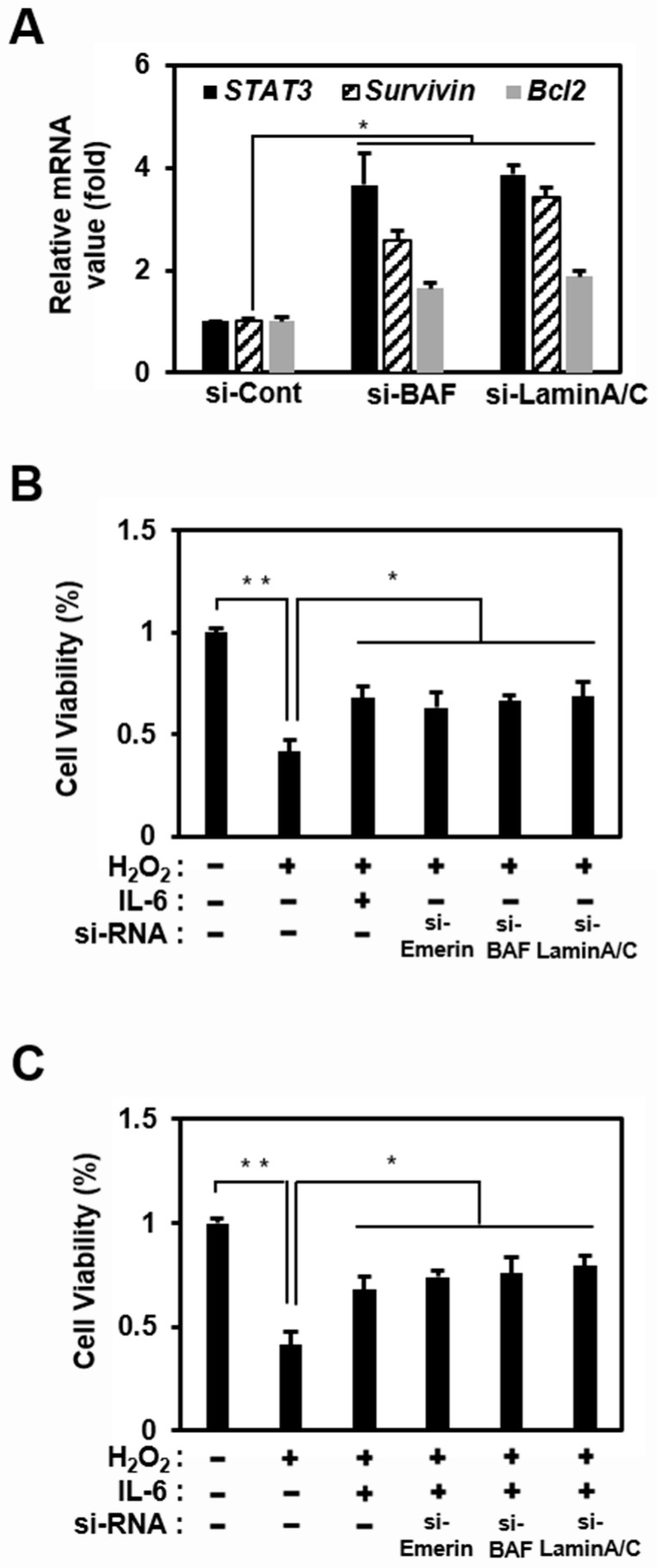
The emerin network modulates STAT3 signaling and survival in HeLa cells. (**A**) HeLa cells were treated with siRNA (100 nM) against *BAF*, *Lamin A/C*, or control for 48 h in 6-well plates. Total RNA was isolated and subjected to qRT-PCR analysis. Data were normalized to GAPDH expression. The results are presented the mean ± SD of three independent experiments performed in triplicate. * *p* < 0.05. (**B**,**C**) HeLa cells were treated with siRNA (100 nM) against *Emerin, BAF*, or *lamin A/C* in 6-well plates for 36 h. HeLa cells were treated with or without IL-6 (40 ng/mL) for 4 h in 6-well plates. Then, HeLa cells were treated H_2_O_2_ (500 μM) for 18 h. Cell viability was measured using the CCK-8 assay in 24-well plates. The results are presented as the mean ± SD of three independent experiments performed in triplicate. * *p* < 0.05, ** *p* < 0.01.

**Figure 4 ijms-22-06669-f004:**
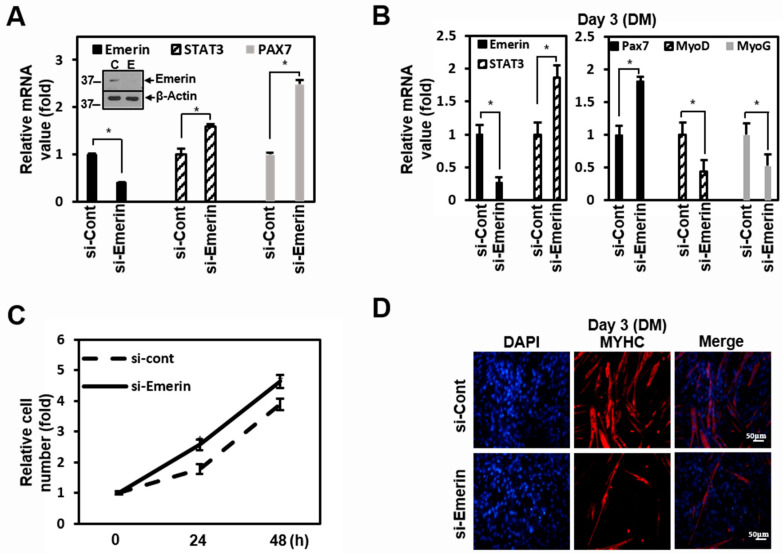
Emerin regulates muscle cell proliferation through STAT3 signaling. (**A**) C2C12 cells were treated with siRNA (100 nM) against *Emerin* or control for 48 h in 6-well plates. The total RNA was isolated and subjected to a qRT-PCR analysis. Data were normalized to *β-Actin*. Treated cells were lysed and subjected to western blotting with antibodies against emerin and β-actin (inner panel). The results are presented as the mean ± SD of three independent experiments performed in triplicate. * *p* < 0.05. (**B**) C2C12 cells were treated with siRNA (100 nM) against *Emerin* or control for 48 h in 6-well plates. After 48 h, cells were treated with differentiation media (DM) for 3 days. The total RNA was isolated and subjected to qRT-PCR analysis. Data were normalized to *β-Actin*. The results are presented as the mean ± SD of three independent experiments performed in triplicate. * *p* < 0.05. (**C**) C2C12 cells were treated with siRNA (100 nM) against *Emerin* or control for 24 or 48 h in 6-well plates. Cells were fixed each time and the relative cell number was counted at each time point. The results are presented as the mean ± SD of three independent experiments performed in triplicate. (**D**) C2C12 cells were treated with siRNA (100 nM) against *Emerin* or control for 48 h in 6-well plates. After 48 h, cells were treated with differentiation media (DM) for 3 days. Immunocytochemistry image stained with anti-MYHC antibody from emerin-depleted C2C12 cells at day 3 of differentiation. DAPI (blue) was used to visualize nucleus. Scale bar = 50 μm.

**Figure 5 ijms-22-06669-f005:**
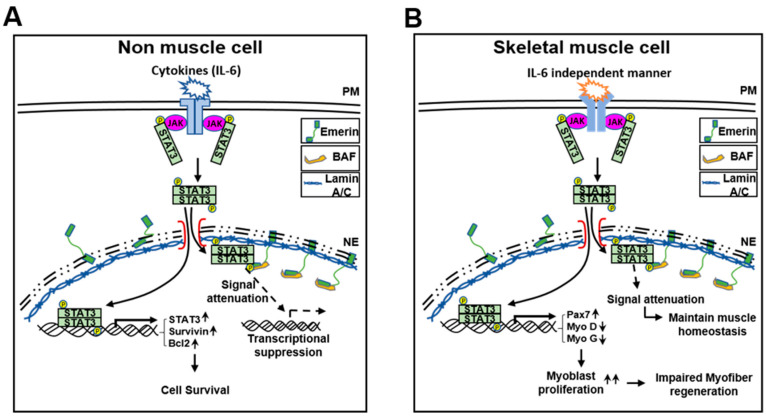
A schematic model describing the role of emerin in the STAT3 signaling pathway. (**A**,**B**) Emerin regulates STAT3 transcriptional activity on non-muscle cells or skeletal muscle cells.

## Data Availability

The data that support the findings of this study are available from the corresponding author upon reasonable request.

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
