# Peer review of "Emerin Represses STAT3 Signaling through Nuclear Membrane-Based Spatial Control"

_ijms, 2021, doi:10.3390/ijms22136669_

Round 1

Reviewer 1 Report

Byoungsun Lee and co-authors presented a work about the role of Emerin in repressing STAT3 transcriptional activity through its sequestration to the nuclear lamina protein scaffold.

The experimental data were obtained by analyzing the transcriptional activity of STAT3 (gene reporter assay) and changes of STAT3 and STAT3-target genes (Bcl2 and Survivin) expression, in cultured cells following up- or downregulation of Emerin. Moreover, IP with WT and deletion mutants of HA-Emerin has been performed to more support the existence of its direct interaction with STAT3 (FLAG-STAT3).

The description of the materials and methods requires a thorough revision, as many details have not been reported. For example, cloning procedures (primers, restriction sites, gene sequence ID) are not reported. IP procedure is not described.

The data presented are convincing only in part, in particular, the IP experiments performed through the overexpression of Emerin mutants. Conversely, discrepancies are found on the gene expression data. Overall, more convinging experiments are required to support the hypothesis and conclusions of the authors.

-STAT3 expression seem to be negatively regulated by Emerin (Fig.1A, 1B, 1D). The same conclusion can be deduced from the experiments performed in C2C12 cells (Fig.4). This hypothesis is further supported by gene reporter assay data (Fig 1E). By contrast, the total level of STAT3 seem to be unchanged following HA-Emerin overexpression (Fig.1F, right panel). How the authors explain this evident discrepancy? Gene expression data must be completed with Western blotting experiments. By analyzing the level of STAT3 (and other proteins) on the same blot, the results are more convincing, especially upon normalization with a housekeeping protein and their representation in graphs.

-Perhaps, In Fig. 1F the blot of HA-epitope should be reported to evaluate the level of HA-Emerin overexpression. Changes in protein levels should be reported in RNA interference experiments.

-The authors stated that H2O2 induced apoptosis in HeLa cells, and IL-6 treatment significantly suppressed H2O2-induced cell death. It is difficult to accept this claim with an MTT assay, without any experimental evidence of apoptotic death. Similarly, the authors do not show any data regarding the anti-apoptotic effect following the silencing of STAT3, LaminA / C and BAF.

-Lines 305-307: the authors stated that “…loss of lamin A/C and BAF leads to erroneous localization and loss of function of emerin to the cytoplasm, so upregulated STAT3 signaling may also contribute to cell survival in HeLa cells”. There is insufficient experimental evidence to support this claim.

- Fig. 5, through this figure the reader understands that Emerin can control STAT3 levels in the cytosol, nuclear membrane and nucleoplasm. This is a strong conclusion that requires more convincing experimental evidences which are not reported in the work. To achieve this, the authors should analyze the levels of free STAT3 in the cytosol and nucleoplasm, and of STAT3 bound to the nuclear membrane (see also lines 161-163).

-It is unclear whether data on transcription factor profiling analysis were firstly reported in this study, or were already presented in another study. These data have been discussed (Discussion), but they were not reported in the results. Furthermore, in the introduction, the authors state that a panel of 58 genes have been analyzed, while only 28 genes are indicated in the supplementary data.

-Fig. 1. Change in gene expression has been reported has fold change (fold). Histograms are not reported correctly, because the referring sample is not indicated (value 1)

-STAT3 analysis in si-Emerin cells was reported twice (1B and 1D). Furthermore, the values of STAT3 level are different from each other.

-It is unclear what Figures 2B and 2C indicate, and how they are different.

-As indicated in Author’s instructions, uncropped images of gels, with size markers etc, must be provided in the Supplementary data.

Reviewer 2 Report

The submitted manuscript shows that emerin interacts with phosphorylated STAT3 and emerin downregulation triggers STAT3 and some of its target genes, including genes involved in cell survival an, in myoblasts, in cellular proliferation. The authors further suggest that lamin A/C and BAF are involved in the proposed mechanisms. 

The topic is of great interest and the experimental plan is appropriate.

Unfortunately, the paper is not well written and many sentences are not at all clear.

Moreover, in some pictures (ex. IP blots), labels are confusing.

Pictures showing the immunofluorescence staining of emerin and STAT3 are of low quality and, most importantly, comparison of STAT3 in control with p-STAT3 in the stimulated sample does not make sense (p-STAT3 must be stained in both sample types).

Finally, densitometric analyses are missing for western blots. 

I suggest to extensively revise this interesting manuscript.

Round 2

Reviewer 1 Report

In the revised version, the authors made substantial changes to the text, and improved the presentation of the data.
Overall, the work is interesting, and provides insights into the interpretation of the role of emerin in the regulation of the STAT3 signaling pathway. However, the reading of the text remains somewhat challenging, and for this reason the work of the authors may be underappreciated.
It would therefore be strongly recommended and mandatory the english editing service before its publication.
I did not find the file containing the uncropped images of the blots, so I encourage the authors to insert them as a supplementary figure.

Finally, the  graph shown in figure 1G is incorrect. The authors reported the values compared with the pSTAT3 density value of control (time 0), in which there is no obvious signal. Hence, the pSTAT3 / STAT3 ratio should be zero (instead, the authors reported the arbitrary unit 1). In these cases, values should be reported as absolute optical density of pSTAT3 normalized with respect to optical density of STAT3. 

Reviewer 2 Report

The paper has been greatly improved by the authors. However, some points still need to be modified.

1) English editing is still required. Some sentences are not clear.

2) In Figure 2 B (now improved by showing p-STAT stining in both samples), only mitotic cells are shown in the insets. This does not make sense. Please, avoid the insets. p-STAT3 and emerin co-localization should be shown in interphase cell nuclei.

3) In Figure 2C, immunoprecipitated emerin-HA bands must be shown.

Round 3

Reviewer 1 Report

Dear Authors

Dear Editors

I have no other comments to add for the review of this work.
As already suggested, the work must be revised from the point of view of the clarity of the exposition, and for this it requires an English editing service. In my opinion, only on this condition, the work can be accepted for publication on IJMS.